# *In vitro* organ culture protocol for intact urogenital systems supporting gonadal differentiation

Sarah L. Whiteley[1]*, Clare E. Holleley[2], Arthur Georges[1]

1 Institute for Applied Ecology, University of Canberra, Canberra, Australia, 2 Australian National Wildlife Collection, CSIRO, Canberra, Australia

* sarah.whiteley@canberra.edu.au

## Abstract

Developing *in vitro* protocols for non-model species poses challenges, and yet are essential for advancing molecular biology studies. This is particularly true for sex determination research that relies on being able to functionally demonstrate the role of genes in determining sex and guiding the process of sex differentiation. Reptile species are attractive models for sex determination research as many species display thermolabile sex systems, allowing for exploration of gene-environment interactions. The first *in vitro* gonad culture technique for a turtle was published almost 35 years ago in 1990, but these techniques have seen limited use. This is likely because of challenges inherent to the system, where cultures need to be maintained for prolonged periods for gonadal differentiation to occur. All published techniques for long-term cultures involve removing the gonad from the surrounding mesonephros, which causes issues for proper testes development. Here we present the first protocol developed in the reptile model, *Pogona vitticeps*, that allows the long-term culture of the whole urogenital system supporting differentiation from biopotential gonads to ovaries or testes. Gross morphology is well maintained, and the gonad can be dissected from the mesonephros even after a culture period of up to 19 days. The cultured gonads display sex specific gene expression and morphology. This protocol will facilitate research in sex determination by providing an effective and low-cost alternative to existing protocols and expand capacity for functional manipulation studies in non-model species.

## Introduction

The ability to conduct experiments *in vitro* has many benefits, particularly for molecular biology research. Such systems are well established for mammalian models, however these techniques pose challenges in other systems. Sex determination is an active area of research, and a suite of non-model reptile species with thermolabile

**Data availability statement:** All raw sequencing files used for the organ culture gene expression analysis are available on NCBI under Bioproject PRJNA1287302. The whole gonad transcriptomes used for comparison are available on NCBI under Bioproject PRJNA699086.

**Funding:** Funding for this project was provided by two Discovery Grants led by AG by the Australian Research Council (DP170101147 and DP220101429). Additional funding was provided to SLW by a CSIRO Research Plus Postgraduate Award. The funders had no role in study design, data collection and analysis, decision to publish, or preparation of the manuscript.

**Competing interests:** The authors have declared that no competing interests exist.

sex determination are of particular interest [1]. The ultimate goal in this field is to demonstrate functional roles for genes in sex determination cascades and their regulation of gonadal differentiation. Despite significant activity in the field for decades, fundamental questions remain unanswered, largely because of difficulties in establishing key molecular biology resources in reptiles.

Despite the challenges, progress has been made in recent years in establishing protocols for functional gene manipulation techniques in whole animals, for example CRISPR editing in *Anolis sageri* [2], and gene manipulation studies using lentivirus in *Trachemys scripta* [3]. However, such techniques pose significant technical challenges, and their routine use has not been established. This necessitates the development of robust *in vitro* culture systems to circumvent the technical and ethical challenges associated with manipulations in whole embryos.

Sex determination research rests on being able to examine the complex genetic and morphological processes governing the differentiation of the biopotential gonads to either ovaries or testes. This poses challenges for *in vitro* culture systems as this process may take weeks (depending on temperature), and differentiation must be sufficiently supported in culture so that the gonad can be clearly identified as an ovary or testis either morphologically, or by gene expression profiles. While no *in vitro* culture system can ever completely recapitulate the environment experienced *in vivo*, the goal is to develop techniques able to sufficiently capture the biological complexity and overcome obstacles present in the *in vivo* system.

These difficulties may explain why, since the publication of the first gonad culture protocol for a reptile in 1990 [4], only seven other studies have since used *in vitro* culture methods (Table 1), all involving just two turtle species (*Lepidochelys olivacea* and *Trachemys scripta*). Egg injection experiments have seen broader use, though they present significant challenges as well, in particular high mortality. For examples, see Ge et al., 2018 where mortality following injection of a lentiviral construct at stage 13 was approximately 50–80% by stage 21. In *P. vitticeps*, injection of antioxidants into eggs caused 41–45% mortality [5]. High mortality creates additional logistical, statistical, and ethical challenges, and certain interventions are embryonic lethal, so can only be successfully executed within a culture system, or may have low efficacy [6]. Injection also raises uncertainty as to the dose received by the embryo in eggs with substantial yolk. Although there are limitations inherent to in vitro culture systems, in such instances they present the only viable alternative.

As outlined in Table 1, the published methods for *in vitro* reptile organ culture are broadly similar, at least in part to having been frequently adapted from each other from two original culture methods for mouse. They can be divided into two groups; those that use Biopore membranes (the majority, originally adapted from a mouse protocol by Taketo and Koide, 1981), and agar slab approaches (only used in two publications, originally adapted from a mouse protocol by Martineau et al., 1997).

An important detail to note is that all previously published protocols culture gonads that have been isolated from the surrounding mesonephros. This prevailing preference of researchers to isolate the gonads likely originates from Shoemaker-Daly et al., (2010), who referenced a personal communication that whole urogenital systems

**Table 1. Summary of published organ culture techniques for reptiles.**

| Species | Culture substrate | Pore size | Culture medium | Culture period | Validation techniques | Year | Reference | Notes |
|---|---|---|---|---|---|---|---|---|
| *Lepidochelys olivacea* | Biopore membrane (Nucleopore Corporation) | 1μM | L15 medium without serum, supplemented with 0.06M of NaCl | 10 days | Light microscopy (Karnovsky solution) | 1990 | [4] | Adapted from protocol developed for mouse by [7] in 1981 |
| *Lepidochelys olivacea* | Low-protein binding Biopore membrane (Millipore) | 0.4 μM | L15 medium, 0.16% NaCl. 10% serum from male or female hatchlings | Up to 13 days | IHC for Sox9 | 2001 | [8] | Hatchling derived serum was not treated to remove steroid hormone. Noted that morphology was improved by use of this type of membrane |
| *Trachemys scripta* | Biopore Millicell membrane (Millipore) | 0.4 μM | L15 medium (with L-Glutamine and phenol red), 10% FBS (unstripped), 0.2% Anti-Anti. The authors noted that phenol red is a weak estrogen mimic and hormones in FBS influenced male development. Subsequently phenol red free L15 and charcoal-stripped FBS was used | Up to 20 days | Histology (H&E), qPCR (*Foxl2, Dmrt1, Sox9, Mis*), Whole-Mount ISH | 2010 | [9] | The authors noted that development was slower in culture, and sex cord formation in testes was poor. First to describe electroporation of plasmid construct in culture to generate mosaic misexpression of *Sox9* |
| *Lepidochelys olivacea* | Biopore membrane | | L15 medium, followed methods by [8] | 72 hours | IHC (Sox9), qPCR (*Sox9, Amh, CYP19A1*) | 2013 | [10] | First study to apply RNAi for *Sox9* in culture |
| *Trachemys scripta* | Agar (1.5% agar in L15) | NA | L15, 10% FBS (charcoal stripped), 50 μg/ml Ampicillin, 1.25 μg/ml Fungizone | Up to 10 days | qPCR (*Wnt4*), IHC (Aromatase, Sox9, Beta-catenin) | 2013 | [11] | Adapted from Martineau et al. 1997 mouse protocol [12] |
| *Trachemys scripta* | Agar (1.5% agar in L15) | NA | L15, 10% FBS (charcoal stripped), 50 μg/ml Ampicillin, 1.25 μg/ml Fungizone | 8 days | IHC (Sox9, Beta Catenin) | 2014 | [13] | Adapted from Mork and Capel 2013, Martineau et al. 1997 |
| *Trachemys scripta* | Low protein binding biopore membrane (Millipore) | 0.4 μM | L15, charcoal stripped FBS, 0.2% pen-strep | Up to 30 days | Histology, IHC (Sox9), qPCR (*Dmrt1, Sox9, CYP19A1, Foxl2*) | 2017 | [14] | Included lentiviral electroporation for DMRT1 knockdown |
| *Trachemys scripta* | Millicell membrane inserts (Millipore) | Not specified | L–15 (no phenol red), 10% FBS (not specified if charcoal stripped), 1 x Anti-Anti, 50 μg/ml Ampicillin | 24 hours | IHC (Gata4), qPCR (*Dmrt1*) | 2020 | [15] | Only study to culture whole UGS, but only for a short period |

do not survive in culture. However, Weber et al., (2020) successfully cultured the whole urogenital system *in vitro*, to assess the effect of a pSTAT3 inhibitor on *Kdm6b* regulation, but only maintained live cultures for 24 hours.

While the isolated gonads tend to perform well in culture, typically displaying expected patterns of sex gene and protein expression, removing them from the supporting mesonephros can impact morphology and growth rate. It also removes the gonads from the hypothalamic-pituitary-adrenal (HPA) axis, which may influence gonad growth and development. As noted by Shoemaker-Daly et al., (2010), gonads developing *in ovo* develop a round shape, whereas in the course of the culture period, the *in vitro* cultured tissues became flattened giving the tissue a more oval shape. *In vitro* gonads developed a less defined medulla and cortex region, and seminiferous tubules were not distinguishable in males. They also noted that morphological development was slower in cultured gonads, contributing to a lengthy culture period (20 days). This is attributed to the removal of the gonads from the surrounding mesonephric tissues. Moreno-Mendoza, Harley and Merchant-Larios, (2001) also noted altered morphology between *in ovo* and *in vitro* gonads, suggesting this is caused by culturing on a Biopore membrane. The mesonephros may provide structural support to gonads in culture, which may improve overall morphology. The mesonephros also supports sex cord formation in the testes, so keeping the urogenital system intact for culture would likely improve testes morphology [9,16]. While culture of the whole HPA axis is impossible, the culture of the intact urogenital system at least includes the adrenal, which may enhance gonad growth and development.

Currently, no organ culture protocols exist for squamates, nor are there protocols for any species allowing the culture of the intact urogenital system for a prolonged period (>24 hours) allowing gonadal differentiation to occur. To address this gap, here we present a protocol developed in the model species, *Pogona vitticeps*, that can maintain the whole urogenital system for up to 19 days. Importantly, the cultured gonads undergo differentiation, and display sex specific characteristics at the morphological and gene expression level. Gross morphology is well maintained such that the gonads can be removed from the mesonephros at the end of the culture period, improving utility for downstream applications such as gene expression analysis.

## Materials and methods

The protocol described in this peer-reviewed article is published on protocols.io (https://www.protocols.io/view/in-vitro-organ-culture-of-intact-urogenital-system-kqdg3qxj1v25/v1) and is included for printing as supporting information file 1 with this article.

## Expected results

**Sample collection and staging.**  All embryonic materials were obtained from eggs laid in the captive breeding colony of *Pogona vitticeps* held at the University of Canberra. All procedures were conducted with approval from the University of Canberra's Animal Ethics Committee.

*P. vitticeps* displays temperature induced sex reversal, where genetic ZZ males develop as females at high temperatures (>32°C). At lower temperatures (typically 28°C) development proceeds in accordance with genotype, where ZZ individuals develop as males and ZW individuals develop as females. After lay, eggs were collected and incubated until developmental stage 6 when they were explanted [17]. ZZ eggs incubated 28°C produce males, while incubation at 36°C which typically produces 96% sex reversal of the ZZ genotype to female [18,19]. We used these conditions to validate that both male and females (including sex reversal) sex differentiation occurs in accordance with these patterns in organ culture.

Previous work has characterised the developmental stages [17] and timing of gonad differentiation in *P. vitticeps* [20]. This staging information is required before starting organ culture experiments in a new species. A balance must be struck between capturing the relevant periods of gonad development and minimising the culture period to avoid aberrant cell behaviour. This must be optimised for each new species depending on their characteristics.

## Method benchmarking

We initially tested culturing on Biopore membranes following previous protocols, choosing 0.4μM pore size as that was the most used (Table 1) Nunc polycarbonate cell culture inserts were used in 6-well plates (0.4μM pore, Thermo Scientific, Cat: 140640). Whole urogenital systems were explanted at stage 4 or stage 6 (bipotential gonads) and cultured until they would be the equivalent of stage 12 (differentiated) depending on incubation temperature. Both 28°C and 36°C incubations were trialled to determine that sex differentiation can proceed as normal, and to establish that temperature induced sex reversal can occur in culture.

As had been noted in previous studies, we found that gross morphology was poorly maintained in all cultures grown on Biopore membranes (Fig 1B,C). Additionally, the organs adhered to the membrane making them difficult to remove for downstream applications without damaging the tissue. Histological examination also found that gonad morphology was poor in females (Fig 1D), and the sex cords in testes were not clearly differentiated (Fig 1E, F). This technique also posed technical challenges for any gene expression sequencing applications as the whole urogenital system must be analysed because insufficient organ integrity means that it is impossible to dissect the gonads from the mesonephros.

To overcome these challenges, we then trialled the culture of whole urogenital systems using agar slabs, adapting the methods outlined by Mork and Capel, (2013). We explanted at stage 6 to keep the culture duration to a minimum, and tested both 28°C and 36°C incubation temperatures (n = 32). We found that the agar slab method produced far superior results. Gross organ morphology was well maintained, such that the gonads were clearly distinguishable from the surrounding kidney tissue, even after the longest culture period of 19 days at 28°C (Fig 1A). This allows the gonads to be dissected out for use in downstream sequencing applications, greatly improving the quality of the results. Importantly, we observed that temperature sex reversal occurs in culture (Fig 1G) and the agar method produces differentiated testes with clear seminiferous tubules (Fig 1H, I), which has previously been difficult to achieve [9].

## Gene expression validation

We verified that the cultured organs recapitulated expected expression patterns in both sex specific and temperature responsive genes previously established for *P. vitticeps* [21,22]. Organs were explanted from stage 6 embryos (laid by ZZf sex reversed mothers, a ZZ x ZZ cross) and cultured for 9 days at 36°C to generate sex reversed females, and 19 days at 28°C to generate concordant ZZm males. Gonads were dissected from the surrounding mesonephros and prepared for RNA sequencing in accordance with procedures described in [21]. Raw count and expression files used for analysis are provided in S2 and S3 Files respectively.

We assessed differential gene expression between 36 ZZf and 28 ZZm *in vitro* organ cultures (S4 File). To assess the similarities or differences in the differential gene expression patterns between 36 ZZf (n = 3) and 28 ZZm groups (n = 3), we performed the same analysis for whole gonads (previously published data in [21,22; S5 File]. We expect the *in vitro* cultures will not completely recapitulate the expression patterns of gonads developing *in situ*, as no culture system can ever capture the biological complexities present in an intact and normally developing embryo. However, we do expect there should be sex specific differences in gene expression patterns, especially given that we know ovaries and testes are distinguishable morphologically (Fig 1). We also expect that specific hallmarks of sex reversal that have been identified in *in situ* gonads will be present in the 36 ZZf *in vitro* cultures.

Differential gene expression analysis between 28 ZZm and 36 ZZf *in vitro* organ cultures showed that well characterised male specific genes *DMRTB1, DMRT1, NR5A1, AMH, GADD45G* were upregulated in 28 ZZm samples (Fig 2, S4 File). *AMH* and *DMRT1* are also differentially expressed between 28 ZZm and 36 ZZf in *in situ* stage 12 gonads, but *NR5A1* is not (Fig 2, S4 File). Bone morphogenic family genes *BMP3*, which is linked to testis development in chicken [23], and *BMP4*, a Sertoli cell marker [24], were also upregulated in *in vitro* 28 ZZm gonads. Hormone synthesis genes *HSD17B7, HSD3B7*, and *HSD17B3* required for testosterone synthesis [25] were upregulated alongside *STAR*. *DLL1, LHX9, PTGER1*. *GATA3* and *BMP7* were upregulated, two genes that have been associated with both sexes. One gene

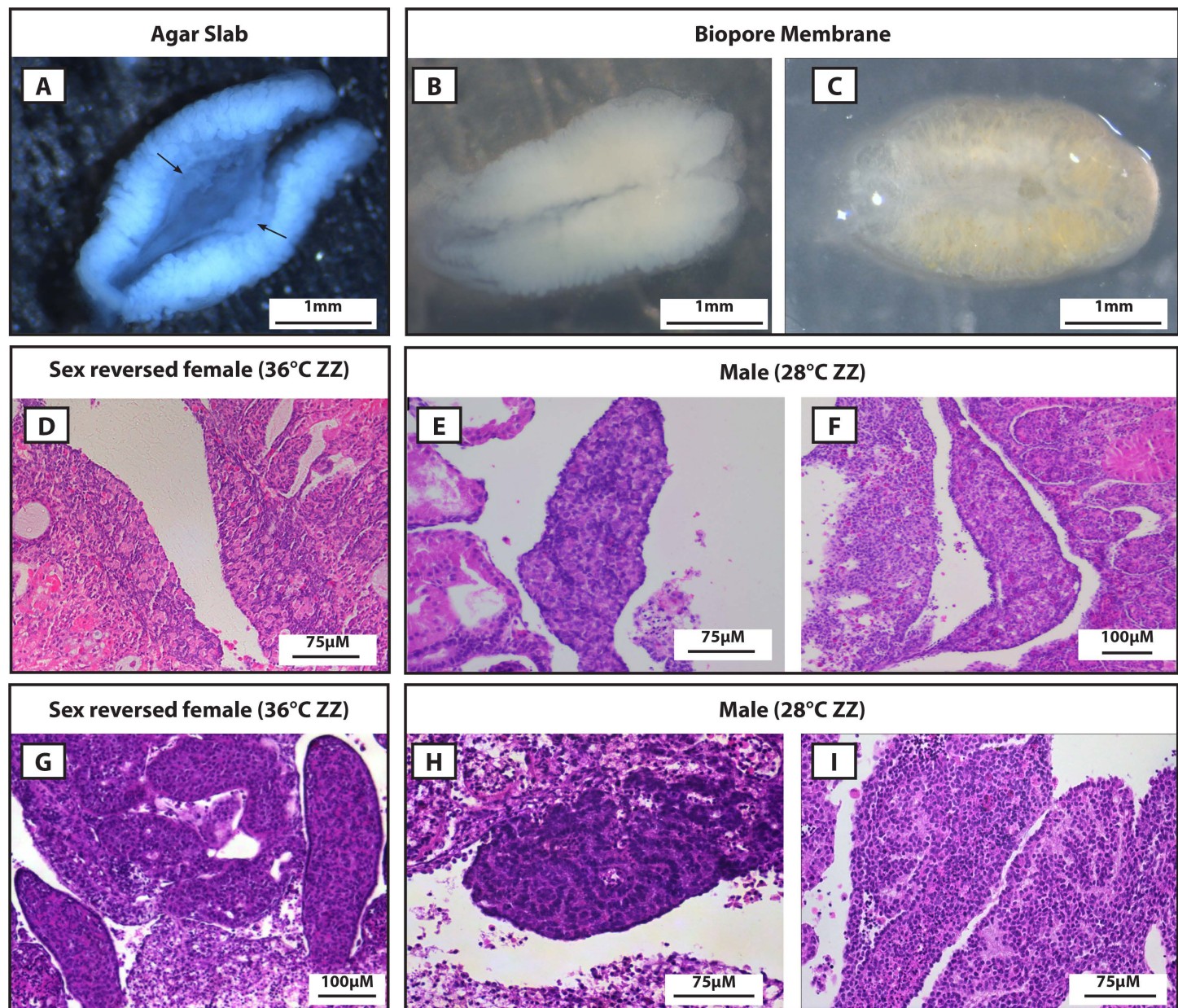

**Fig 1. Comparison of gross morphology (A-C) and histology (D-I) between agar slab (A, D, G) and Biopore membrane (B, C, E, F, H, I) organ culture techniques in *Pogona vitticeps*.** A) Gross morphology of whole urogenital system cultured for 19 days at 28°C showing gonads (black arrows) clearly distinguishable from the surrounding mesonephros tissue. B and C) Gross morphology of organs cultured on a Biopore membrane showing the gonads are not distinguishable from the surrounding mesonephros tissue. D) Histology (H & E stained) section of ZZf sex reversed female incubated at 36°C on a Biopore membrane. The gonads are not clearly separated from the mesonephros and although primordial germ cells have proliferated, the typical ovarian characteristics of a well-defined and proliferating medulla and degrading cortex are not clear. E and F) Histology (H & E stained) sections of ZZm males incubated at 28°C on a Biopore membrane. Similarly to the gonad in D, germ cells have proliferated but typical testes characteristics of a degenerate cortex and medulla with well-defined seminiferous tubules are not obvious. All sections shown in D, E and F, it is not possible to confidently assign a testes or ovary phenotype as the morphology is too ambiguous. G) Histology (H & E stained) section of ZZf sex reversed female incubated at 36°C on an agar slab. The gonads are well defined from the surrounding mesonephros. While the ovarian characteristics are not as clear as for an ovary developing *in ovo*, the cortex layer is well-defined and the cells in the medulla are disordered. H and I) Histology (H & E stained) sections of ZZm males incubated at 28°C on an agar slab. Seminiferous tubules are difficult to achieve in culture, however as shown in H it is possible for them to clearly develop as is typical for testes developing *in ovo*. I shows testes with less well-developed seminiferous tubules, but that are still obvious enough that it could be distinguished from an ovary.

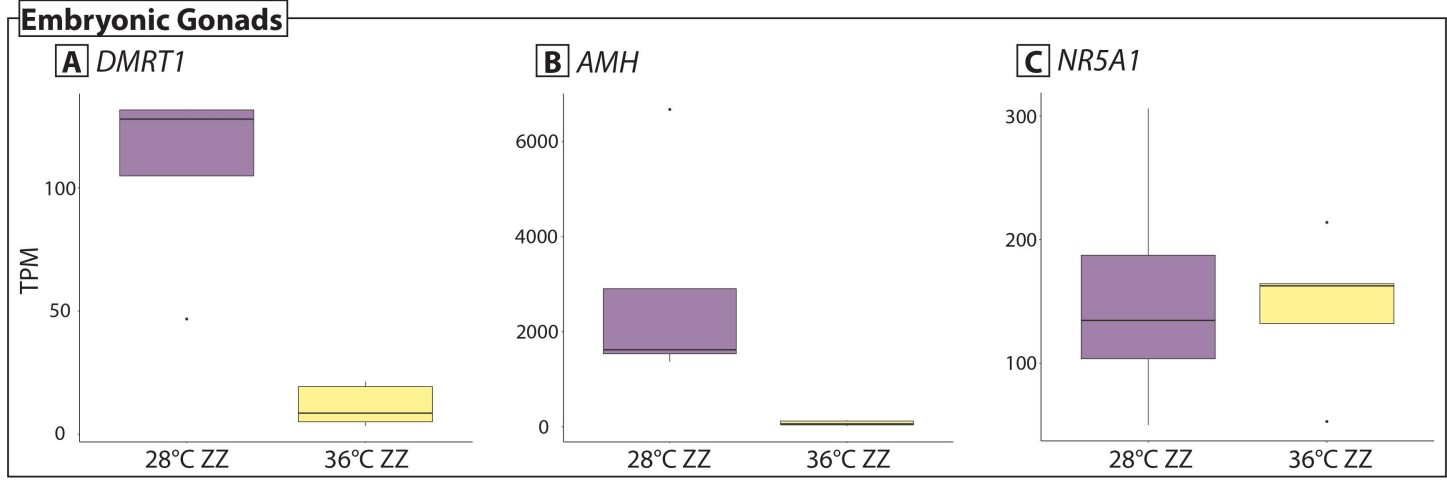

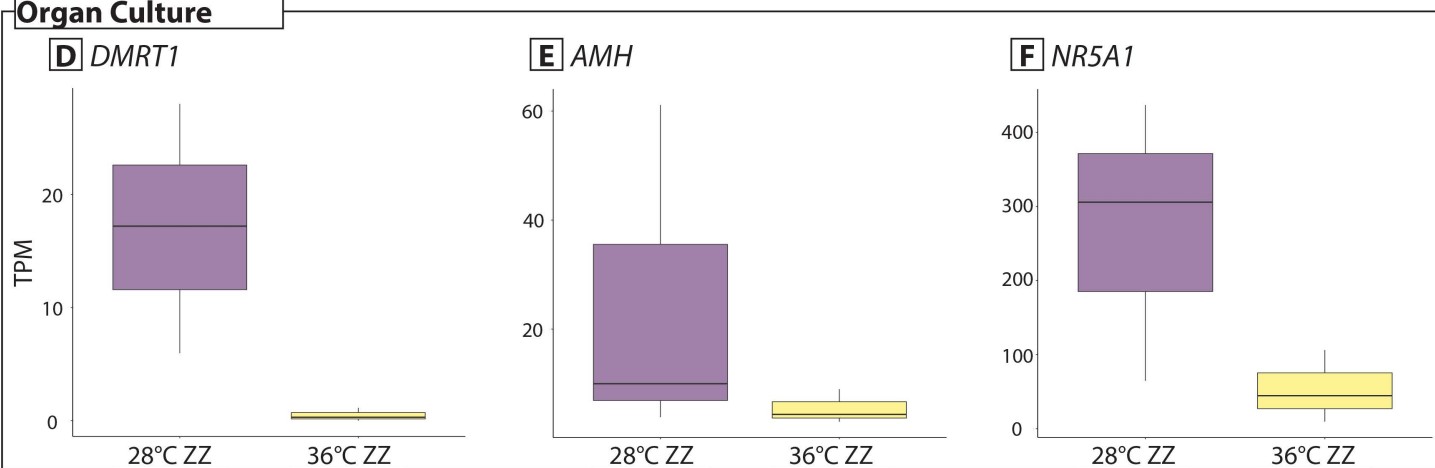

**Fig 2. Gene expression (TPM, transcripts per million) of male associated genes.** Expression of *DMRT1, AMH,* and *NR5A1* from whole embryonic gonads at stage 12 that developed *in situ* (data from [21,22].Panels A-C), and gonads isolated from organ cultures (panels D-F).

typically associated with female development was upregulated, however the log-fold change was low (*RSPO2;* log-fold change = 3, *p* = 0.02; S4 File). In the 28 ZZm *in situ* gonads, *HSD17B3* was upregulated but none of the other HSD family genes were, and *STAR* was upregulated in 36 ZZf *in situ* gonads. Despite some differences in the gene expression patterns between *in situ* and *in vitro* gonads, overall the cultured organs displayed clear gene expression patterns expected for males.

In the 36 ZZf *in vitro* cultured gonads (expected to be all sex reversed females), sex specific gene expression trends were not as pronounced as those observed in the 28ZZ males. This may be because high temperatures mask some of the sex-specific gene expression patterns, and there appears to be differences in the timing of gene expression compared to gonads developing *in situ*. In the 36 ZZf *in situ* gonads, there were some unusual trends. *CYP19A1* and *FOXL2* were significantly upregulated, as expected for females, however strongly male associated genes were also upregulated, including *SRD5A2,* which converts testosterone to the more potent dihydrotestosterone, and *AMHR2* (Fig 3, S4 File).

However, the strong down-regulation of male related genes (as outlined above), and upregulation of some female related genes does support that sex reversal can occur in the organ culture system. Vitellogenin-2-like was one of the

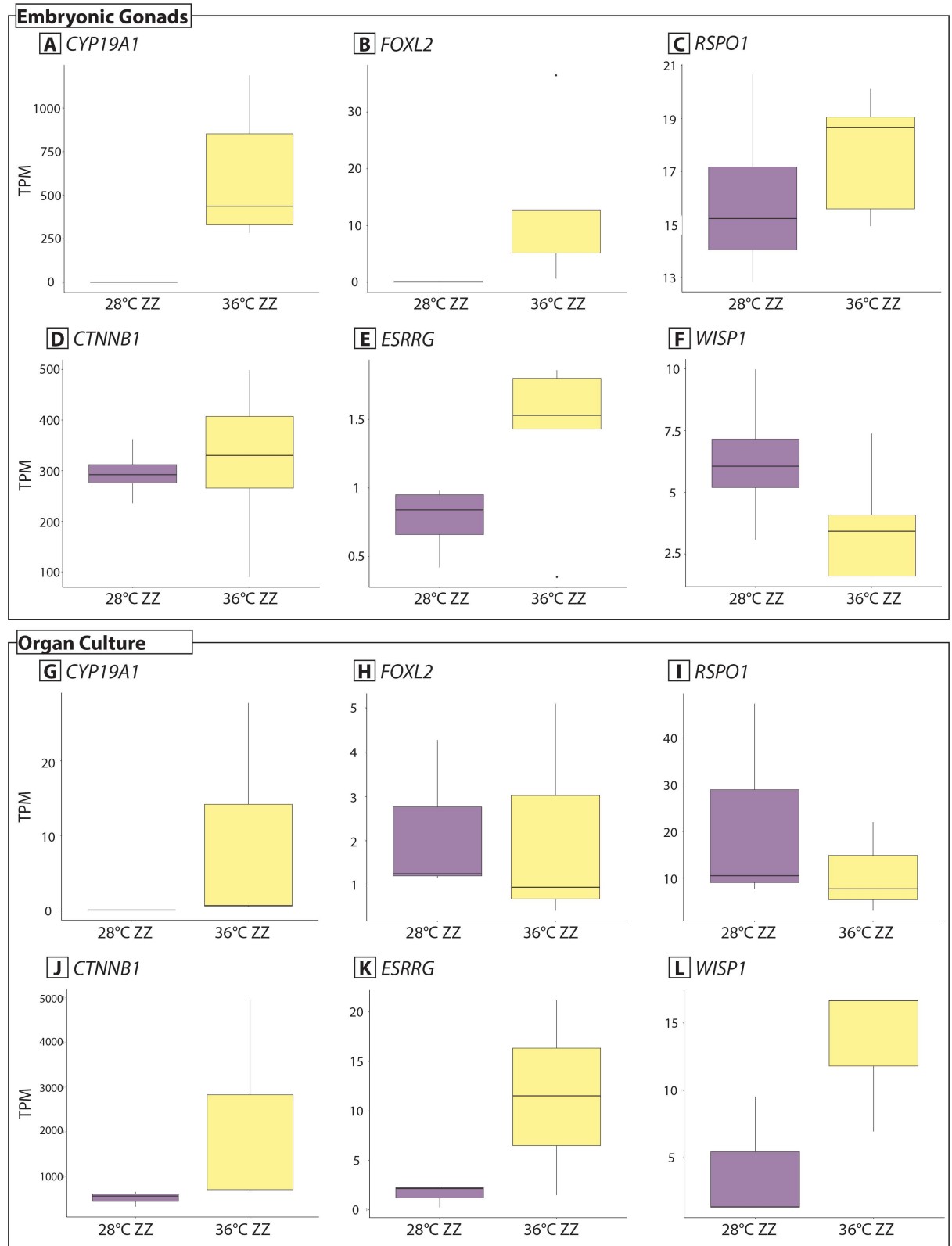

**Fig 3. Gene expression (TPM, transcripts per million) of female associated genes.** Expression of *CYP19A1, FOXL2, RSPO1, CTNNB1, ESRRF,* and *WISP1* from whole embryonic gonads at stage 12 that developed *in situ* (data from [21,22].Panels G-L), and gonads isolated from organ cultures (panels D-F).

most highly expressed genes, alongside hormone synthesis genes *HSD17B2* and *ESRRG*. Other female development associated genes *WNT10B, LHX8, PAX8, WISP1* were also upregulated [21,23]; Fig 3). Female genes *CYP19A1, RPOS1, FOXL2, and CTNNB1* were lowly expressed in culture (Fig 3). Given that the cultures do differentiate clear ovarian characteristics (Fig 1G), these expression trends suggest that these genes must be expressed at higher levels prior to when the gonads were sampled in order to support ovarian development. Notably, *RSPO1* and *CTNNB1* are not differentially expressed in *in situ* gonads.

Chromatin modifiers *JARID2* and *KDM6B*, and thermosensitive gene *CIRBP*, were highly upregulated, and based on our previous work, this is a strong signal of sex reversal [21] Fig 4). These patterns were recapitulated in gonads developing *in situ* where they were upregulated in 36 ZZf (S5 File). Sex reversal in organ culture is also supported by gonadal morphology (Fig 1G).

Even for *in situ* gonads, expression patterns can be complicated, and with the addition of the temperature influences, and be complicated even further. Because of this, care should be taken if using gene expression profiling in general, but particularly for *in vitro* organ cultures. This is especially important for species with thermosensitive sex determination

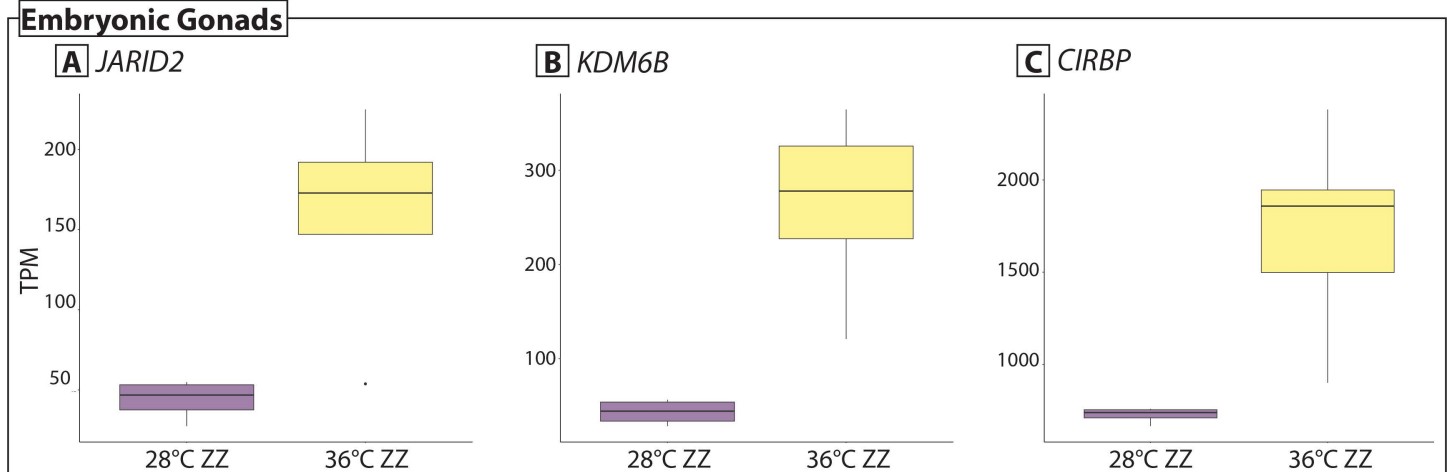

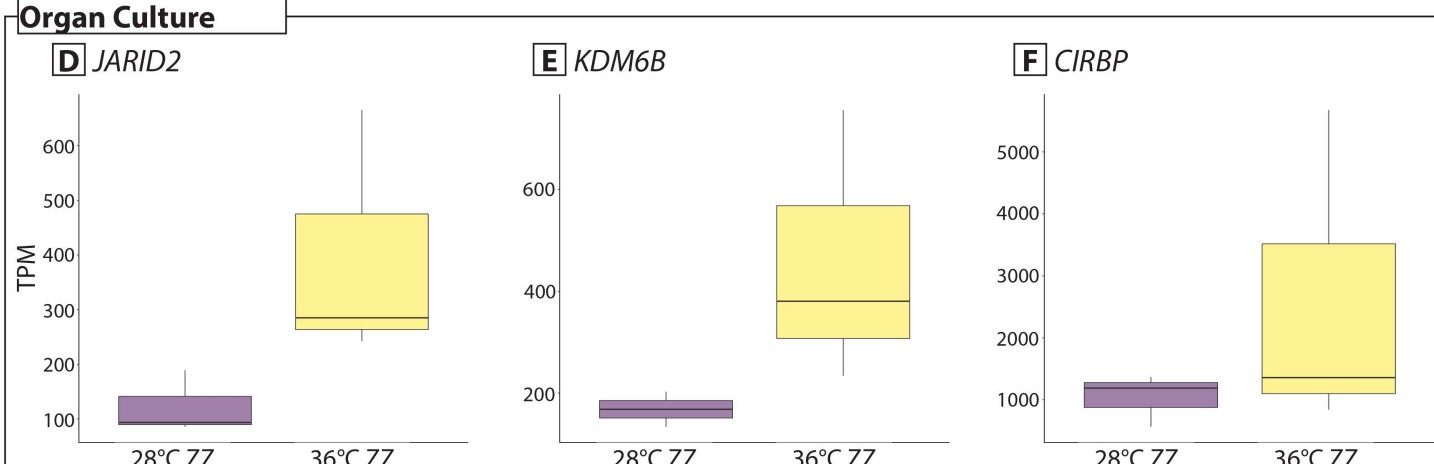

**Fig 4. Gene expression (TPM, transcripts per million) of sex reversal associated genes.** Expression of *JARID2, KDM6B,* and *CIRBP* from whole embryonic gonads at stage 12 that developed *in situ* (data from [21,22].Panels A-F), and gonads isolated from organ cultures (panels D-F).

systems where the influence of temperature on sex differentiation is fundamentally linked. Whole transcriptome sequencing provides capacity to better understand this variation, or uncover unexpected patterns in expression profiles. Many other sequencing techniques are available for validating expression profiles, for example single cell sequencing, but are often cost prohibitive. Other cost effective alternatives, such a qPCR, should be carefully implemented as the expression profiles of a smaller number of genes may be more difficult to interpret. Combining any sequencing techniques with other morphology techniques, such and histology and immunohistochemistry, represents a balanced strategy to give a more complete overview of cellular characteristics and gene expression patterns in culture. Other assays assessing hormonal function could also be considered depending on research requirements.

## Supporting information

**S1 File. Step-by-step protocol, also available on protocols.io.**
(PDF)

**S2 File. Raw count files for sex reversed ZZf females incubated at 36°C (n = 3) and ZZm males incubated at 28°C (n = 3) cultured in vitro.** Analysis was conducted using methods described in (21). The sample IDs correspond to incubation temperature (28 or 36), genotype (ZZ), replicate (1, 2 or 3).
(CSV)

**S3 File. Raw expression files (TPM, transcripts per million) for sex reversed ZZf females incubated at 36°C (n = 3) and ZZm males incubated at 28°C (n = 3) cultured in vitro.** Analysis was conducted using methods described in (21). The sample IDs correspond to incubation temperature (28 or 36), genotype (ZZ), replicate (1, 2 or 3).
(CSV)

**S4 File. Results from differential gene expression analysis between sex reversed ZZf females incubated at 36°C (n = 3) and ZZm males incubated at 28°C (n = 3) cultured *in vitro*.** Analysis was conducted using methods described in [21]. Log-fold change threshold of −1, 1 and $p$ value cut-off of 0.05 have been applied.
(XLSX)

**S5 File. Results from differential gene expression analysis between sex reversed ZZf females incubated at 36°C (n = 3) and ZZm males incubated at 28°C (n = 3) from gonads developed *in situ*.** Analysis was conducted using methods described in [21] using data from [21,22]. Log-fold change threshold of −1, 1 and $p$ value cut-off of 0.05 have been applied.
(XLSX)

## Acknowledgments

The testing and now frequent use of this organ culture method came about because of the COVID-19 pandemic restricting the availability of the Biopore membranes. We are grateful for this unintended, yet ultimately positive outcome. We thank Chelsea Steele and her animal husbandry team at the University of Canberra for their care of the *P. vitticeps* animal colony that supports our research. We acknowledge the contributions of Blanche Capel and her team in being the first to adopt this agar slab method in reptiles, and for providing the methodological foundation which we based this protocol on. We also thank members of Team Pogona, past and present, for their support throughout the development of this protocol. Associated content: https://www.protocols.io/view/in-vitro-organ-culture-of-intact-urogenital-system-kqdg3qxj1v25/v1.

## Author contributions

**Conceptualization:** Sarah L. Whiteley, Arthur Georges.

**Data curation:** Sarah L. Whiteley.

**Formal analysis:** Sarah L. Whiteley.

**Funding acquisition:** Clare E. Holleley, Arthur Georges.

**Investigation:** Sarah L. Whiteley.

**Methodology:** Sarah L. Whiteley, Clare E. Holleley.

**Project administration:** Clare E. Holleley, Arthur Georges.

**Resources:** Sarah L. Whiteley, Clare E. Holleley, Arthur Georges.

**Supervision:** Clare E. Holleley, Arthur Georges.

**Validation:** Sarah L. Whiteley.

**Writing – original draft:** Sarah L. Whiteley.

**Writing – review & editing:** Sarah L. Whiteley, Clare E. Holleley, Arthur Georges.

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
