## [Decision Letter · Decision Letter 0]

PONE-D-24-57396In vitro organ culture protocol for intact urogenital systems supporting gonadal differentiationPLOS ONE

Dear Dr. Whiteley,

Thank you for submitting your manuscript to PLOS ONE. After careful consideration, we feel that it has merit but does not fully meet PLOS ONE’s publication criteria as it currently stands. Therefore, we invite you to submit a revised version of the manuscript that addresses the points raised during the review process.

We look forward to receiving your revised manuscript.

Kind regards,

Rajakumar Anbazhagan

Academic Editor

PLOS ONE

**Journal Requirements:**

1. When submitting your revision, we need you to address these additional requirements. Please ensure that your manuscript meets PLOS ONE's style requirements, including those for file naming. The PLOS ONE style templates can be found at https://journals.plos.org/plosone/s/file?id=wjVg/PLOSOne_formatting_sample_main_body.pdf and https://journals.plos.org/plosone/s/file?id=ba62/PLOSOne_formatting_sample_title_authors_affiliations.pdf 2. Thank you for stating the following financial disclosure: Funding for this project was provided by two Discovery Grants led by AG by the Australian Research Council (DP170101147 and DP220101429). Additional funding was provided to SLW by a CSIRO Research Plus Postgraduate Award.  Please state what role the funders took in the study.  If the funders had no role, please state: "The funders had no role in study design, data collection and analysis, decision to publish, or preparation of the manuscript." If this statement is not correct you must amend it as needed. Please include this amended Role of Funder statement in your cover letter; we will change the online submission form on your behalf. 3. In the online submission form, you indicated that All sequencing data has been deposited with NCBI and accession numbers are available on request. All PLOS journals now require all data underlying the findings described in their manuscript to be freely available to other researchers, either a. In a public repository, b. Within the manuscript itself, or c. Uploaded as supplementary information.This policy applies to all data except where public deposition would breach compliance with the protocol approved by your research ethics board. If your data cannot be made publicly available for ethical or legal reasons (e.g., public availability would compromise patient privacy), please explain your reasons on resubmission and your exemption request will be escalated for approval. 4. When completing the data availability statement of the submission form, you indicated that you will make your data available on acceptance. We strongly recommend all authors decide on a data sharing plan before acceptance, as the process can be lengthy and hold up publication timelines. Please note that, though access restrictions are acceptable now, your entire data will need to be made freely accessible if your manuscript is accepted for publication. This policy applies to all data except where public deposition would breach compliance with the protocol approved by your research ethics board. If you are unable to adhere to our open data policy, please kindly revise your statement to explain your reasoning and we will seek the editor's input on an exemption. Please be assured that, once you have provided your new statement, the assessment of your exemption will not hold up the peer review process. 5. Your ethics statement should only appear in the Methods section of your manuscript. If your ethics statement is written in any section besides the Methods, please move it to the Methods section and delete it from any other section. Please ensure that your ethics statement is included in your manuscript, as the ethics statement entered into the online submission form will not be published alongside your manuscript.

Reviewers' comments:

Reviewer's Responses to Questions

**Comments to the Author**

1. Does the manuscript report a protocol which is of utility to the research community and adds value to the published literature?

Reviewer #1: Yes

Reviewer #2: Yes

2. Has the protocol been described in sufficient detail?

To answer this question, please click the link to protocols.io in the Materials and Methods section of the manuscript (if a link has been provided) or consult the step-by-step protocol in the Supporting Information files.

The step-by-step protocol should contain sufficient detail for another researcher to be able to reproduce all experiments and analyses.

Reviewer #1: Yes

Reviewer #2: Partly

3. Does the protocol describe a validated method?

Reviewer #1: Yes

Reviewer #2: Yes

4. If the manuscript contains new data, have the authors made this data fully available?

Reviewer #1: Yes

Reviewer #2: Yes

**5. Is the article presented in an intelligible fashion and written in standard English?**

Reviewer #1: Yes

Reviewer #2: Yes

6. Review Comments to the Author

**Reviewer #1:**  The current article describes an in vitro organ culture protocol involving agar slab for intact urogenital systems in the reptile model, Pogona vitticeps supporting differentiation of biopotential gonads into either ovaries or testes. The protocol delivered superior results over the existing protocols with respect to integrity of gross organ morphology, showed sex specific gene expression and temperature induction of sex reversal during culture. Overall it is providing a reliable method for studying gonadal differentiation in non-model species.

**Reviewer #2: ** This In-vitro organ culture protocol for intact urogenital systems is significant. Unlike traditional genetic studies, which rely on whole-animal experiments, this method allows direct functional analysis of gene-environment interactions. It is substantial for reptiles, where temperature-dependent sex determination (TSD) is a key factor, enabling controlled studies.

A step forward, but not without its challenges. These include species-specific adaptations, the need for external supplementation of systemic factors, and potential developmental deviations in vitro. However, the future is promising, with potential improvements such as integrating hormonal signaling, optimizing culture conditions for broader species applicability, and exploring co-culture systems to better mimic in vivo environments. The journey towards a more comprehensive understanding continues.

The authors need to discuss the points below to mark.

1. In vivo, gonadal development is influenced by systemic factors such as hormones from the hypothalamus-pituitary-gonadal (HPG) axis. This in vitro system, while valuable, lacks these interactions, which may lead to incomplete or altered differentiation patterns. It's important to acknowledge these limitations and consider external supplementation with hormones or signaling molecules to mimic in vivo conditions fully.

2. Even though the gross morphology of the gonads is preserved, subtle changes in differentiation could still occur due to the artificial culture environment. The absence of vascularization and mechanical forces in vivo may affect the development of specific cell types or tissue organization. The impact of in vitro conditions on long-term gonadal function and reproductive potential remains unclear.

3. The study acknowledges that no in vitro system can fully replicate the biological complexity of in situ development. However, it does not sufficiently discuss the potential impact of missing systemic factors, such as endocrine signals or maternal influences, on gonadal differentiation and gene expression. The absence of these factors in the in vitro system may lead to incomplete or altered differentiation patterns, which could affect the expression of sex-specific genes and the overall reproductive potential of the cultured gonads.

4. The study highlights that Biopore membranes fail to maintain gonad morphology, but the agar slab method does not fully replicate in vivo development. The extent to which these morphological differences affect downstream analyses (e.g., gene expression) is not deeply explored. The expression of sex-specific genes in cultured gonads does not fully match that observed in in situ development, particularly in 36°C-induced sex-reversed females. The study suggests that high temperatures may mask sex-specific gene expression, but alternative explanations should be considered, such as differences in developmental timing or stress responses in vitro. These factors could potentially influence gene expression and should be further investigated.

5. While the study presents morphological and gene expression evidence of sex reversal, inconsistencies in gene expression trends (e.g., up regulation of some male genes in sex-reversed females) raise questions. Additional functional validation (e.g., hormonal assays or long-term differentiation studies) would strengthen the conclusions.

6. The inability to cleanly separate gonads from the mesonephros in Biopore cultures limits the specificity of gene expression data. Even in agar slab cultures, residual mesonephric tissue could influence RNA sequencing results. To address this, the study could further evaluate the influence of residual mesonephric tissue on RNA sequencing results by comparing single-cell RNA sequencing data, which would provide a more detailed and accurate picture of gene expression in cultured gonads.

7. The study does not clearly state the number of biological replicates used for gene expression analysis. Given the complexity of temperature-induced sex reversal, a larger sample size would help determine whether observed trends are biologically meaningful or due to individual variability.

8. While agar slabs performed better than Biopore membranes, other 3D culture techniques, such as extracellular matrix-based hydrogels, might provide better structural support and improve morphological and molecular outcomes. The study does not explore these possibilities.

9. The study optimizes culture conditions based on previous P. vitticeps staging but does not test whether earlier or later developmental stages might yield better results. A more detailed assessment of optimal culture windows could improve the model.

7. PLOS authors have the option to publish the peer review history of their article (what does this mean? ). If published, this will include your full peer review and any attached files.

**Do you want your identity to be public for this peer review?** For information about this choice, including consent withdrawal, please see our Privacy Policy .

Reviewer #1: No

Reviewer #2: No

---

## [Author Response · Author response to Decision Letter 1]

23 May 2025

Manuscript PONE-D-24-57396: In vitro organ culture protocol for intact urogenital systems supporting gonadal differentiation

Response to reviewers

We thank the Editor and the two Reviewers for taking the time to consider our manuscript for publication in PLOS One. The manuscript was well received by Reviewer 1 who didn’t provide any suggestions for the manuscript, stating that the protocol provides “a reliable method for studying gonadal differentiation in non-model species.”

We have responded in detail to the comments from Reviewer 2 in BLUE, and actions we have taken to address these comments are in RED. In the amended manuscript the amendments have been made using tracked changes as requested by the editorial team, along with a clean revised version. All referenced line numbers correspond to the tracked changes version of the manuscript.

Reviewer 2 Response

We appreciate the thoroughness of the response from Reviewer 2. We note that the majority of the comments share a theme in that there are limitations of our present method that are inherent to ex ovo culture systems. It is not possible to ever perfectly replicate the in vivo environment in culture.

We acknowledged this in the original submission on lines 187-189, where we stated that “we expect the in vitro cultures will not completely recapitulate the expression patterns of gonads developing in situ, as no culture system can ever capture the biological complexities present in an intact and normally developing embryo.”

This includes the absence of any signals that may originate from the hypothalamus (comment 1) and other endocrine signalling factors (comment 3) that may mean that there are “subtle changes” in the organ culture (comment 2).

ACTION: In order to address the overarching concerns central to these comments, we have added the following lines to the revised manuscript:

“While no in vitro culture system can ever completely recapitulate the environment experienced in vivo, the goal is to develop techniques able to sufficiently capture the biological complexity and overcome obstacles present in the in vivo system.” (lines 57-60).

“Although there are limitations inherent to in vitro culture systems, in such instances they present the only viable alternative for experimentation.” (lines 71-72).

The other concern raised by Reviewer 2 consistent across three comments was that additional validation may shed more light onto the efficacy of the culture method. This includes hormonal assays (comment 5), single cell RNA-sequencing (comment 6), and matrix-based hydrogels (comment 8).

We agree that there are many options available for validating and optimising in vitro culture protocols, as described in lines 241-249 of the original submission. We suggest the utility of whole transcriptome sequencing but also acknowledge other more cost-effective approaches like qPCR, and other morphology based techniques alongside histology, such as immunohistochemistry.

One of the major advantages of our method is its simplicity and low cost (mentioned in lines 35-36), making it accessible to all researchers compared to techniques that rely on expensive consumables like biopore membranes and hydrogel matrixes.

ACTION: To highlight the current limitations of our presented method, we have added the following lines and suggest additional avenues for optimisation of the technique, we have added the following lines:

“Many other sequencing techniques are available for validating expression profiles, for example single cell sequencing, but are often cost prohibitive.” (lines 248-249).

“Other assays assessing hormonal function could also be considered depending on research requirements.” (lines 254-255).

Other comments from Reviewer 2

1. In vivo, gonadal development is influenced by systemic factors such as hormones from the hypothalamus-pituitary-gonadal (HPG) axis. This in vitro system, while valuable, lacks these interactions, which may lead to incomplete or altered differentiation patterns. It's important to acknowledge these limitations and consider external supplementation with hormones or signaling molecules to mimic in vivo conditions fully.

We thank Reviewer 2 for their comment and agree that any in vitro system has inherent limitations and will never be able to fully recapitulate the in vivo environment, as addressed in response to comments above.

We also note that the organ culture is not fully separated from the entire HPA (hypothalamus-pituitary-adrenal) axis because the intact UGS is explanted such that the adrenal is still present. This is an advantage to previous methods where the gonad was cultured in isolation, so was completely removed from the whole axis. We also note that there is limited evidence for the role of the HPA axis in reptile sex determination (reviewed in Castelli et al, Biological Reviews, 2020).

Any signalling molecules potentially originating from the HPA axis are poorly characterised in reptiles, so it would be difficult to determine any necessary supplementation to the culture medium. Additionally, as outlined in this paper, the gonads of multiple reptile species can differentiate in vitro suggesting that any influence from the HPA axis is not critical to this process.

ACTION: To address this suggestion from Reviewer 2, we have added the following statement to lines 90-91: “It also removes the gonads from the hypothalamic-pituitary-adrenal (HPA) axis, which may influence gonad growth and development”.

We also added the following on lines 102-103: “While culture of the whole HPA axis is impossible, the culture of the intact urogenital system at least includes the adrenal, which may enhance gonad growth and development in culture.”

4. The study highlights that Biopore membranes fail to maintain gonad morphology, but the agar slab method does not fully replicate in vivo development. The extent to which these morphological differences affect downstream analyses (e.g., gene expression) is not deeply explored. The expression of sex-specific genes in cultured gonads does not fully match that observed in in situ development, particularly in 36°C-induced sex-reversed females. The study suggests that high temperatures may mask sex-specific gene expression, but alternative explanations should be considered, such as differences in developmental timing or stress responses in vitro. These factors could potentially influence gene expression and should be further investigated.

Throughout the manuscript we have been careful to never claim that our agar slab method fully replicates the in vivo environment. Indeed, we have drawn attention to the fact this is not possible in several parts of the manuscript, and have made further additions to the revised paper in response to other comments from Reviewer 2 in relation to this.

We do agree with Reviewer 2 that temperature complicates the gene expression patterns observed in the 36°C sex reversed females, as stated in the original submission: “This may be because high temperatures mask some of the sex-specific gene expression patterns, and there appears to be differences in the timing of gene expression compared to gonads developing in situ.” (lines 214-216).

On lines 241-243 we also stated that: “Even for in situ gonads, expression patterns can be complicated, and with the addition of the temperature influences, and be complicated even further. Because of this, care should be taken if using gene expression profiling in general, but particularly for in vitro organ cultures.”

ACTION: To further highlight the complicating effects of temperature, we have added the following: “This is especially important for species with thermosensitive sex determination systems where the influence of temperature on sex differentiation is fundamentally linked.” (lines 244-246).

7. The study does not clearly state the number of biological replicates used for gene expression analysis. Given the complexity of temperature-induced sex reversal, a larger sample size would help determine whether observed trends are biologically meaningful or due to individual variability.

ACTION: We have included sample sizes for the different groups on lines 172, 189, 262, 266. The sample sizes are comparable to previous studies, particularly for transcriptomics (see Whiteley et al., 2021, 2022 for examples).

9. The study optimizes culture conditions based on previous P. vitticeps staging but does not test whether earlier or later developmental stages might yield better results. A more detailed assessment of optimal culture windows could improve the model.

As stated on lines 133-135 of the original submission, we did test the influence of explanting tissue earlier in development (stage 4). We noted on lines 16-169 that we chose stage 6 to “keep the culture duration to a minimum”. Similarly, we did not explant later in development as previous work on the species showed that differentiation can occur as early as stage 8, and we did not want to risk explantation when the gonad was committed to a sexual fate (Whiteley et al., Scientific Reports, 2018).

---

## [Decision Letter · Decision Letter 1]

In vitro organ culture protocol for intact urogenital systems supporting gonadal differentiation

PONE-D-24-57396R1

Dear Dr. Whiteley,

We’re pleased to inform you that your manuscript has been judged scientifically suitable for publication and will be formally accepted for publication once it meets all outstanding technical requirements.

Kind regards,

Academic Editor

PLOS ONE

Additional Editor Comments (optional):

Reviewers' comments:

Reviewer's Responses to Questions

**Comments to the Author**

1. Does the manuscript report a protocol which is of utility to the research community and adds value to the published literature?

Reviewer #1: Yes

Reviewer #2: Yes

2. Has the protocol been described in sufficient detail?

To answer this question, please click the link to protocols.io in the Materials and Methods section of the manuscript (if a link has been provided) or consult the step-by-step protocol in the Supporting Information files.

The step-by-step protocol should contain sufficient detail for another researcher to be able to reproduce all experiments and analyses.

Reviewer #1: Yes

Reviewer #2: Yes

3. Does the protocol describe a validated method?

Reviewer #1: Yes

Reviewer #2: Yes

4. If the manuscript contains new data, have the authors made this data fully available?

Reviewer #1: Yes

Reviewer #2: Yes

**5. Is the article presented in an intelligible fashion and written in standard English?**

Reviewer #1: Yes

Reviewer #2: Yes

6. Review Comments to the Author

Reviewer #1: The authors have well addressed the comments made by reviewers. The authors added sections to acknowledge the limitations of in vitro culture system including their proposed model that those cannot fully recapitulate the environment experienced in vivo. However, the proposed model is an improvement on in vitro culture models in that it cultures the intact urinogenital system and can be an experimental choice for in vitro culture model.

Reviewer #2: The authors have addressed the major concerns thoughtfully, particularly regarding the limitations of in vitro systems and the absence of systemic endocrine inputs. The inclusion of adrenal tissue is noted, though it does not fully replicate HPG/HPA axis signaling. The clarifications added to the manuscript are appreciated. Functional validation (e.g., hormonal assays) and advanced culture platforms (e.g., hydrogels) could further enhance the model’s utility. Additionally, assessing a broader developmental window may optimize outcomes. This study represents a promising and accessible approach to investigating reptilian sex determination. With further refinement, the model could offer significant and hopeful insights into the environmental and molecular mechanisms of gonadal development.

7. PLOS authors have the option to publish the peer review history of their article (what does this mean? ). If published, this will include your full peer review and any attached files.

**Do you want your identity to be public for this peer review?** For information about this choice, including consent withdrawal, please see our Privacy Policy .

Reviewer #1: No

Reviewer #2: No

---

## [Editor Report · Acceptance letter]

PONE-D-24-57396R1

PLOS ONE

Dear Dr. Whiteley,

I'm pleased to inform you that your manuscript has been deemed suitable for publication in PLOS ONE. Congratulations! Your manuscript is now being handed over to our production team.

Kind regards,

on behalf of

Dr. Rajakumar Anbazhagan

Academic Editor

PLOS ONE